# Optical Genome Mapping for Cytogenetic Diagnostics in AML

**DOI:** 10.3390/cancers15061684

**Published:** 2023-03-09

**Authors:** Verena Nilius-Eliliwi, Wanda M. Gerding, Roland Schroers, Huu Phuc Nguyen, Deepak B. Vangala

**Affiliations:** 1Center for Hemato-Oncological Diseases, Department of Medicine, University Hospital Knappschaftskrankenhaus Bochum, Ruhr-University Bochum, 44892 Bochum, Germany; verena.nilius-eliliwi@ruhr-uni-bochum.de (V.N.-E.);; 2Human Genetics, Ruhr-University Bochum, 44801 Bochum, Germany

**Keywords:** AML, Optical Genome Mapping, cytogenetics, whole-genome sequencing, WGS

## Abstract

**Simple Summary:**

Today, the classification of acute myeloid leukemia is mainly based on genetic aberrations found in leukemic cells. Classifying this disease is necessary for exact risk stratification, which, in turn, is relevant for treatment decisions. The genetic diagnostics employed include the detection of single-nucleotide variants as well as larger structural and copy number variants. The latter are currently detected and analyzed through the combination of several methods such as chromosomal banding analysis, fluorescence in situ hybridization, and molecular genetics. Here, we review the current evidence regarding the use of Optical Genome Mapping as a single genome-wide technique for the detection of structural and copy number variations in acute myeloid leukemia.

**Abstract:**

The classification and risk stratification of acute myeloid leukemia (AML) is based on reliable genetic diagnostics. A broad and expanding variety of relevant aberrations are structural variants beyond single-nucleotide variants. Optical Genome Mapping is an unbiased, genome-wide, amplification-free method for the detection of structural variants. In this review, the current knowledge of Optical Genome Mapping (OGM) with regard to diagnostics in hematological malignancies in general, and AML in specific, is summarized. Furthermore, this review focuses on the ability of OGM to expand the use of cytogenetic diagnostics in AML and perhaps even replace older techniques such as chromosomal-banding analysis, fluorescence in situ hybridization, or copy number variation microarrays. Finally, OGM is compared to amplification-based techniques and a brief outlook for future directions is given.

## 1. Introduction

Recently, progress has been made in the management of acute myeloid leukemia (AML). One reason for this is the introduction and approval of an increasing number of targeted treatments, which are not only employed in combination with intensive induction chemotherapy regimens but also as monotherapies for patients who are ineligible for intensive protocols. Furthermore, the refinement of risk stratification has led to more individualized therapies for AML patients. The most commonly used stratification system by the European Leukemia Net (ELN) has been updated in 2022 (Table 1a) [1]. It relies on correct disease classification, which, in turn, is solely based on genetic variants in leukemic cells [2,3]. These genetic variants include single-nucleotide variants (SNVs), such as those in *NPM1*, but also a variety of cytogenetic changes. Thus, the correct and detailed identification of structural variants (SVs) is one of the main objectives in classifying patients with AML.

In this study, we review the current methods of cytogenetic diagnostics and summarize the knowledge on novel techniques with a focus on optical genome mapping (OGM) via the Saphyr^®^ system developed by Bionano Genomics (San Diego, CA, USA) as a novel diagnostic tool and a scientific instrument used in the analysis of AML. We compare the evidence of this technique to that of other modern approaches such as whole-genome sequencing (WGS). Finally, we provide an outlook on a modern diagnostic approach to AML.

## 2. Diagnostic Necessities in AML

Patients with de novo AML, i.e., without previous myeloproliferative or myelodysplastic disorders, often present with an acute to subacute onset of symptoms over a period ranging from a few days to a few weeks. These usually unspecific symptoms, such as fatigue or dyspnea, can be accompanied by other clinical and laboratory findings, of which the most important include changes in full blood count with reduced platelet and hemoglobin levels accompanied by elevated levels of lactate dehydrogenase and uric acid. This combination of patient history, clinical symptoms, and laboratory findings leads to the suspicion of acute leukemia as an underlying cause. From an epidemiological standpoint, the incidence of AML in adults is about three times higher than that of acute lymphoblastic leukemia (ALL) [4].

The initial and most important diagnostic procedure is a quick microscopical evaluation of smear samples of peripheral blood and bone marrow. A blast count above 20% in the bone marrow (depending on the used classification system) confirms the diagnosis of acute leukemia in most cases. Lineage differentiation is usually performed by flow cytometry analysis. Cytologic evaluation and flow cytometry are readily available at most hematological centers. Therefore, a suspected diagnosis of AML can usually be made within hours [5].

The detailed classification of AML is solely based on genetic findings. In this regard, there is a high degree of overlap between the WHO classifications and the International Consensus Classifications (ICC), which were both updated in 2022 [2,6]. These systems label a variety of recurrent genetic abnormalities as AML-defining, either irrespective of the blast count (WHO) or with at least 10% blasts (ICC). On the one hand, these recurrent genetic changes include variants in *NPM1* and *CEBPA* and treatment relevant changes in *FLT3*, *IDH1*, and *IDH2*. Furthermore, mutations in a variety of genes known to be associated with myelodysplastic syndrome (MDS) have been incorporated and include genes such as *ASXL1*, *BCOR*, *EZH2*, *RUNX1*, *SF3B1*, *SRSF2*, *STAG2*, *U2AF1*, and *ZRSR2*. Variants in these genes are typically detected by amplification-based methods, of which the most important are next-generation sequencing (NGS) techniques, either by specific panels or by whole exome/whole-genome-sequencing (WES/WGS) approaches. The ELN recommends testing for an additional set of 20 genes upon diagnosis, which are not (yet) part of the current classification system [1].

Apart from that, there are several AML-relevant gene fusions and rearrangements (Table 1b). For these structural variants (SV), a timely diagnosis can be important and the most important fusion transcripts can be quickly identified by Multiplex-RT-PCR. However, there is a wide range of SVs, such as insertions, deletions, and translocations, that are detected by cytogenetic methods. Furthermore, any aberration in karyotype can be relevant for correct classification and hence essential for treatment decisions. This becomes clear when examining the ELN2022 risk stratification, where any chromosomal aberration not labelled as favorable or adverse leads to an intermediate risk [1]. In terms of therapy for patients who qualify for intensive treatment, this could entail a decision for or against allogeneic hematopoietic stem cell transplantation as consolidation treatment. Recently, in clinical practice, accurate stratification according to genetic criteria has been essential in order to individualize treatment. This development was considered in last year’s versions of the above-mentioned classification systems. Nevertheless, there are still several patients (approx. 15%) who present without any of the currently known aberrations.

## 3. Conventional Cytogenetic Procedures in AML Diagnosis

Classical chromosomal banding analysis (CBA) has been the standard baseline cytogenetic method for decades. It is broadly available and cost-effective. In order to create a karyogram, the chromosome structure must be visible for microscopy, which is possible in the metaphase of the cell division cycle. Therefore, the cells under analysis must be cultured for 2–3 days. Afterwards, the cell cycle is synchronized and the cells are arrested in metaphase, fixed on a slide, and stained, e.g., by trypsin enzyme digestion using Giemsa (GTG). Then, the condensed chromosomes are visible under a light microscope and can be arranged to form a classical karyogram. Visual evaluation allows the experienced examiner to detect numerical and structural aberrations with a resolution limit of around 5–10 megabases (Mb) for constitutional cytogenetic preparations and >10 Mb for leukemia samples [7]. CBA offers the benefit of an unbiased analysis, meaning that any aberration above this threshold is detectable. This contrasts with targeted methods, where only selected regions of interest are specifically analyzed.

However, during cultivation, certain clones of cells might benefit from in vitro conditions, thus resulting in growth artifacts and some selection bias. Furthermore, the minimum number of metaphases that ought to be analyzed is 20. Considering the immense number of cells and potentially different malignant clones, this is a comparatively small number, which may not be representative of the entire sample. With the help of artificial intelligence (AI)-based programs, the time of analysis can be decreased and the number of analyzed metaphases increased [8,9]. Taken together, the method is nevertheless time consuming since a complete workup might require between 7 to 10 days under real-life conditions.

A molecular expansion of CBA is fluorescence in situ hybridization (FISH). This targeted approach lowers the resolution threshold to about 100–200 kb in routine diagnostics and to as much as 5 kb with specific alterations [10,11]. Fluorescent probes that are complementary to the target genomic region are mainly hybridized to interphase DNA. The fluorescent probe binds to the complementary DNA, and the colored signal can be observed using a fluorescence microscope. Several probes of different colors can be combined. This method allows for the visualization of gene disruptions or translocations due to structural rearrangements or numerical changes. This analysis can be performed on interphase nuclei; therefore, the cultivation of cells is not mandatory. Usually, 200 cells are analyzed, which means that FISH is more representative of the actual variant distribution than CBA and even small proportions of malignant clones are detectable [12]. On the other hand, as FISH is a targeted approach, it can only be used for the detection of specific aberrations for which probes are available. For a detailed analysis of a large set of potential SVs, FISH is of limited use, not only because of the cost of each probe but also due to the limited number of commercially available and validated probes. For a whole-genome analysis, the use of multicolor FISH is possible [13,14]. However, due to the high costs incurred because of the necessity of using several probes to address such questions, FISH is usually utilized in routine diagnostics for specific questions only.

Another method for genetic diagnostics in acute leukemias is copy number analysis by microarray (CMA) methodology. This is a genome-wide technique that is restricted to the analysis of copy number changes, i.e., the loss or gain of genetic material deviating from the normal, diploid state. This method is a molecular cytogenetic approach because gross deviations are typed using molecular techniques. In principle, DNA extracted from sample material is used to analyze single nucleotide polymorphisms, being part of the genetic variation, that are typed using hybridization probes over the entire genome (SNP-microarray) followed by bioinformatic analysis. Alternatively, the sample in question can be mixed with a control sample, and deviations are evaluated for this comparative genomic hybridization (array CGH). Generally, about 12–250 copy number variants (CNV) can be found per individual, which can affect 40–460 Mb of the entire genome [15]. The mean detection resolution of this methodology can be down to 50 kb or less depending on the platform and array used as well as the marker density in the region in question. Structural changes such as translocations or inversions without a loss or gain of genetic material cannot be detected. In contrast to CBA and FISH, this method’s sensitivity to detect low-level aberrations in a subset of cells is limited. The advantage of the use of CMA in AML diagnosis is its ability to detect a copy-neutral loss of heterozygosity, which is a prognostic factor in AML [16,17].

By combining CBA, FISH, RT-PCR, and CMA, the majority of SVs can likely be detected. However, apart from the above-mentioned limitations of each individual method, this approach is time consuming and costly. Therefore, cytogenetic diagnostics would benefit from a single, genome-wide approach with high resolution. In this regard, technologies such as OGM or WGS approaches are currently being evaluated.

## 4. Optical Genome Mapping

OGM is a novel, genome-wide method for cytogenetic diagnostics (Figure 1) [18,19]. The method described in this section and the data summarized in this review were exclusively generated via the commercially available Saphyr^®^ system developed by Bionano Genomics (San Diego, CA, USA). This also includes sample workups and data analysis on a platform provided by the company. OGM is based on the analysis of native ultra-high-molecular-weight (UHMW) DNA, i.e., >150 kb. Approximately 1.5 × 10^6^ cells are required to gain enough DNA material. Thus, a small sample of easily accessible EDTA blood or bone marrow is sufficient for analysis. Samples for OGM can be stored at room temperature for 4 days or frozen at ≤−80 °C before further processing. The extraction of UHMW DNA is more time consuming than regular genomic DNA preparation and takes about one day. Next, the DNA is labelled with an enzyme called “Direct label and stain enzyme 1” (DLE-1). DLE-1 adds a green fluorochrome to every region of the DNA, where a six-base-long sequence (CTTAAG) is present. On average, this sequence occurs about 15 times per 100 kb. Additionally, the whole DNA backbone is stained with a blue, fluorescent dye. Labeled DNA is loaded on a chip containing nano-channels, in which the DNA molecules are linearized and stretched by electrophoresis and then run on an instrument (Saphyr^®^, Bionano genomics) that contains the equipment with which to image labelled and linearized DNA molecules via fluorescence microscopy. The pictures taken are analyzed by a software that recognizes the fluorescence pattern (barcode pattern) and assembles the analyzed genome according to the barcode pattern in comparison to a reference genome (Genome Reference Consortium human build 37 (GRCh37)). A canonical gene set (either human genome 19 or 38 (hg19 or hg38)) can be chosen for annotation as a reference genome for human samples.

The software offers two methods of analysis: the de novo pipeline (DNP) and the rare variant pipeline (RVP). With the DNP, the DNA molecules are assembled to form a complete genome, which, in turn, is compared to the reference genome (GRCh37/hg19 or hg38). The DNP offers the advantage of high resolution but the limitation of being incapable of detecting aberrations with a low allele frequency. During RVP analysis, labelling patterns are aligned against the reference genome (GRCh37/hg19 or hg38). Differences of the aligned barcode pattern from the reference genome lead to the creation of a consensus genome map file (CMAP). In turn, the CMAP is realigned with the reference genome; differences in the alignment pattern are then referred to as SVs. Every aberration needs to be detected in three DNA molecules to be called as such. The RVP is more commonly used for the analysis of hematological malignancies as it offers the possibility of detecting the mosaicism of aberrations. CNVs have a minimum length of about 500 kb and are detected by the quantification of DNA molecules in comparison to the average number of all DNA molecules. The platform used for detection is the CN analysis tool, which is a component of both the DNP and RVP [20]. To achieve a valid analysis, specific quality parameters must be reached, namely, a map rate ≥70%, a label density of 14 to 17/100 kb, and an average N50 ≥ 230 kb for molecules ≥ 150 kb. The N50 is a value representing the average length of DNA molecules. The analysis is dependent on the UHMW DNA being of sufficient quality, i.e., DNA molecules of sufficient length. To simplify the analysis, the sample genome can be compared to a database containing 179 genomes of ethnically diverse, healthy individuals, thereby filtering out benign variations. Furthermore, browser-extensible data (BED) files either provided by the platform, e.g., the canonical cancer genes, or that have been custom-made can be employed to quickly find the aberrations of interest in predefined gene sets.

Taken together, a coverage of more than 300× can be achieved. With this degree of coverage, the method reaches a detection limit for mosaicism of at least 5% for SVs and 10% for CNVs/aneuploidies [20]. The resolution threshold is about 0.5 kb to 5 kb for SVs depending on the analysis pipeline and 500 kb for CNVs, which is about 1000–20,000 times higher than that for CBA. Although these metrics support the use of OGM as a single tool for cytogenetic diagnostics, the method has a few intrinsic limitations. The detection of SVs in centromeric and telomeric regions of the genome can be challenging. Here, coverage is limited due to the high number of repetitive sequences and, therefore, lack of labelling (so-called masked regions). Furthermore, sometimes cases of hyper- or hypodiploidy are missed. As the software calculates a relative number of gene copies, it is prone to mistakes if the entire genome has a hyper- or hypodiploid state.

In contrast, some additional benefits of OGM are its omission of DNA amplification and cell cultivation procedures. Therefore, the variant allele fraction (VAF) detected by OGM is representative of the sample. This opens up the possibility of detecting different types of malignant clones, which are also undergoing evolution over the course of the disease, when analyzing samples from different time points.

## 5. Evidence Regarding the Use of OGM in Diagnosis of Hematological Malignancies

In recent years, OGM has been used for various indications. For hematological malignancies, there is increasing evidence of its usefulness, especially regarding the growing need for genetic diagnostics. Most of the published studies have focused on the comparison of OGM and standard cytogenetic diagnostics. The latter included CBA virtually all analyzed samples and, in most works with a focus on AML, FISH and CNV-microarrays where needed as by treating physicians’ choice. This likely represents the real-life diagnostic approach in most centers in the developed world. The studies of OGM in hematological diagnostics primarily focused on the concordance of results regarding cytogenetic findings through OGM compared to routine diagnostics. Furthermore, all studies looked for additional information that could be acquired through OGM. This additional information was twofold: on the one hand, unclear findings such as marker chromosomes or unclear complex karyotypes, usually due to the low resolution of CBA, could be resolved. On the other hand, FISH and CMAs were not utilized in all samples but were used as indicated by treating physicians and wherever established in routine diagnostics. Thus, additional information also included SVs detected by OGM, that, in theory, might be observed via methods such as FISH, which would not be part of routine diagnostics for these specific SVs. As FISH probes are usually only utilized for predefined aberrations, these variants would be missed by standard cytogenetic diagnostics.

In 2021, Neveling et al. compared OGM to standard diagnostics (CBA, FISH, and CMA) in 52 samples of various hematological malignancies (11 AML, 19 MDS, 1 ALL, 8 CLL, 3 CML, 5 Lymphoma, 2 myeloproliferative Neoplasia (MPN), 1 multiple Myeloma (MM), and 1 T-PLL). In this cohort, OGM detected the variants described by standard diagnostics in 50/52 (96%) cases. Furthermore, OGM led to a more detailed understanding of the underlying aberrations, especially in complex cases, with regard to cytogenetics [21].

In a similar approach, Sahajpal et al. confirmed these results. They analyzed 59 hematological samples (18 AML, 12 MDS, 15 CLL, 3 MPN, 2 CML, 6 MM, and 3 Lymphomas) in comparison to standard diagnostics (CBA, FISH, and CMA) and demonstrated a 99% concordance rate with additional information detected in several cases [22]. The additional information mostly consisted of redefined and refined karyotypes. Especially complex karyotypes with multiple gains and losses and derivative or marker chromosomes could be presented in more detail. In most cases, the origins of the marker chromosomes were uncovered.

A couple of groups focused their work on distinct hematological entities, mostly acute leukemias. So far, Puiggros and colleagues are the only ones that have provided data on CLL. They analyzed 42 patients, reaching a 90% concordance rate with standard diagnostics (CBA, FISH, and CMA) and additional information in 55% of patients. Furthermore, they showed that the classification of patients according to the obtained OGM results was more precise in predicting the time to first treatment compared to standard diagnostics, which is a clinically relevant prognostic marker in CLL [23].

For ALL, in a small cohort of 10 samples from children and adult patients, Lestringant et al. found a 90% concordance rate (standard diagnostics: CBA, FISH, SNP-Array, and RT-MLPA (reverse transcription multiplex ligation-dependent probe amplification)) [24].

Lühmann et al. showed a 100% concordance rate (standard diagnostics: CBA, FISH, SNP-array, and RNA-sequencing) in 12 childhood ALL cases. In this study, the refinement of karyotypes was possible in 75% of cases [25].

Rack et al. analyzed 41 pediatric and adult ALL cases. Standard diagnostics included CBA, FISH, SNP-array, MLPA, and RT-PCR in this study [26]. The authors detected a concordance rate of 100% and a gain of additional cytogenetic information in 75% of cases. So far, analyses of 63 ALL cases have been published. OGM was clearly not inferior in comparison to current cytogenetic diagnostics. In contrast, OGM was able to detect novel variants, especially gene fusions, and to redefine the karyogram, especially in cases of complex aberrations. However, as ploidy changes play an important role in ALL, currently, CBA cannot be fully replaced by OGM with respect to diagnostics for ALL.

## 6. Optical Genome Mapping in AML—Concordance with Standard Diagnostics

As mentioned above, the need for more precise genetic diagnostics has become evident for AML in recent years. We addressed this need by implementing OGM in parallel with our routine diagnostics in AML and MDS. So far, 35 AML and 7 MDS patients have been analyzed at our hematological tertiary center [27,28]. The median age of our study population was 61 years with 59% male patients. According to ELN2017 classification, 17% of our AML patients were stratified in the favorable risk group, whereas 37% belonged to the intermediate and 42% to the adverse risk groups, respectively. All patients underwent standard-of-care diagnostic procedures including blood and bone marrow sampling with cytological, histopathological, and flow cytometry analysis. Furthermore, panel sequencing regarding actionable targets was performed. All patient samples underwent routine cytogenetic diagnostics, which included successful CBA for 32/35 AML patients and FISH (8/35), CNV microarrays (6/35), and Multiplex RT-PCR (28/35) for gene fusions according to treating physicians’ indications. In parallel, OGM was performed, as described above (Figure 2). To simplify the read-outs and to evaluate this method for routine use, we primarily analyzed a predefined AML-relevant gene set focusing on 185 genes. This gene set was based on classification-relevant aberrations and potential genes of interest as described by the Cancer Genomics Consortium [29]. We found a 91% concordance rate for the AML samples with additional information in 64%. The additional information mainly consisted of further aberrations, e.g., *KMT2A*-PTD in two cases, *NF1*-deletions in two cases, a *BCOR* deletion, and rearrangements of *NUP98* and *MECOM* in one case, respectively. Furthermore, in three patients, OGM allowed for the description of a karyotype, which was not possible after routine CBA. In 12 cases, the karyotypes were refined. In one case, rather than a normal karyotype, a complex karyotype with several aberrations leading to deletions in *NF1*, *TP53*, and *ETV6* was present. This did not change risk classification for this patient because of a simultaneously existing risk defining *FLT3*-ITD. In another case, additional information obtained by OGM led to a change in the patient’s risk stratification from adverse to favorable, which meant that alloHSCT was not performed in this case.

The largest study regarding OGM for AML was published by Levy et al., including 100 AML cases. The standard diagnostics applied were CBA for all cases, FISH for 19, and CMA for 3 cases. The authors found a 100% concordance rate for SVs with a VAF > 5% and a concordance rate of 95% irrespective of the VAF. In 13% of samples, additional information could be detected by OGM. In 12% of cases, the information gained by OGM led to changes in classification and hence therapeutic decision making [30].

Balducci et al. published a study with 41 AML and 27 MDS patients. In 88% of the AML cases, concordance with standard diagnostics was reached [31]. Notably, apart from CBA, the percentage of patients analyzed with FISH was higher than in the study by Levy and co-workers (63% vs. 19%). In 41% of cases, additional information was obtained, and in 5%, classifications or treatment strategies were changed because of additional information obtained by OGM.

Suttorp et al. analyzed 24 pediatric AML and mixed-lineage leukemia cases. As for the adult population, one patient was re-classified, leading to changes in treatment. For two patients, an MRD marker could be identified [32].

In summary, so far 205 cases of adult patients with AML have been published with availability of clinical data. OGM showed a concordance rate of about 90% with CBA and extended cytogenetic diagnostics. The proportion of patients where additional information had been added through OGM varied between 13% and 64%. This is probably due to the different definitions that the respective groups had chosen in order to classify which pieces of information were labelled “additional”. OGM resulted in a change in risk stratification in 3–12% of the study population (Table 2).

The limitations of OGM in nearly all studies were the detection of aberrations with a low VAF (<5%). In our samples, we could see that the threshold for the detection of mosaicism could be reduced to a VAF of 1–2% with a higher coverage of 600× (unpublished data). Furthermore, ploidy changes were not detected in some cases; for instance, in our study, 2/3 of the missed aberrations were trisomies. Regarding this issue, we saw an improvement with the latest software version (Bionano access 1.7.1.1). However, of the three ploidy changes (monosomy—eleven; trisomy—eight; and double-minutes chromosomes—seven) observed via CBA and missed by OGM in the study by Suttorp et al., one could not be verified by FISH, and another was likely to be an artifact as it did not occur in follow-up CBAs.

Several of the AML studies included MDS patients. For MDS only, Yang at al. delivered the largest data set, which contained 101 patients. They combined OGM with NGS using an 81-gene panel. The data obtained were complementary. In three cases, OGM failed to detect aberrations, all of which were present in two to three metaphases, only [33].

In our study with 42 patients, 7 were diagnosed with MDS. In these cases, concordance was demonstrated in 83%, with additional information added in 50% of cases [27,28].

Balducci et al. published a study with 68 patients, 27 of whom were MDS patients, reaching a 94% concordance rate and gaining additional information in 19% of cases. In 22%, a change in IPSS-R was obtained [31]. Notably, these MDS cases were heterogenic, including only cases with excessive blasts in our work, but also other subtypes of MDS in other cohorts. However, especially with respect to MDS, due to the impact of SVs, OGM could be an important diagnostic tool in the future.

## 7. Cytogenomics via Amplification-Based Methods in Comparison to OGM

OGM is a method for the detection of aberrations at a chromosome-wide level with a resolution down to 0.5 kb, as described above. Single-nucleotide variants (SNV) cannot be seen. In clinical practice, amplification-based methods such as NGS or PCR-based methods are essential, as a broad variety of SNVs are relevant for classification. Apart from mere risk stratification, today, a growing number of SNVs can be addressed by targeted treatments, e.g., inhibitors in case of activating mutations in *IDH1*, *IDH2*, or *FLT3*. Thus, all cytogenetic tools described above need to be augmented by amplification-based techniques. Usually, this work-up includes panel sequencing or RT-PCR analysis.

Utilizing genome-wide sequencing could lead to a real “one-stop-shop” procedure for genetic diagnostics, including the detection of all SVs and SNVs. Some studies have analyzed AML samples with WGS or WES to uncover information about the genetic drivers in, for example, a relapse situation [34,35].

Duncavage et al. performed the largest study so far comparing WGS to standard cytogenetics in myeloid malignancies [36]. WGS analysis was performed as an automated approach that reported aberrations in 40 genes, CNVs larger than 5 Mb, and 612 recurrent SVs. A coverage of 60× was targeted. With this automated analysis, the median turnaround time was 5.1 days in weekly batches (equaling the minimal turn-around time for OGM). In 94% of cases, additional methods apart from WGS were not required (apart from PCR analysis for *FLT3*-ITD, which was included for every sample). This shows that if WGS is combined with an automated analysis focusing on a predefined gene panel, bioinformatics can be simplified, and hands-on time for trained personnel can be reduced. In the study cohort, 175 AML samples were included. A 100% concordance rate with standard cytogenetics and additional information in 17% of cases were obtained. For 102 patient samples, WGS was compared to a targeted sequencing panel that aimed for 500× coverage. The concordance rates were 85% for SNVs and 92% for insertions and deletions (indels), respectively. The missed aberrations occurred either with low VAF or were placed in areas with low coverage.

For 68 AML patients, Duncavage et al. compared the diagnostic yield of WGS and a PCR assay for *FLT3* mutations with a standard diagnostic approach consisting of CBA, FISH, and a targeted sequencing panel. In this instance, additional information was delivered for 25% of cases. In 15%, this led to a change in ELN risk classification. Thus, WGS indeed offers a quick and thorough diagnostic approach. However, the rate of undetected SNVs is in the same range as undetected SVs for OGM. Furthermore, WGS is dependent on reliable capacities for bioinformatics and is probably more costly regarding hardware and personnel. Focusing on cytogenetic variants, a head-to-head comparison between amplification-free genome mapping and WGS has not yet been performed.

Notably, apart from OGM, there are other platforms focusing on genome-wide SV analysis, such as the Oxford Nanopore platform for which diagnostic studies are currently underway [37].

## 8. OGM as a Tool for Detecting Novel Variants

OGM’s high concordance with standard cytogenomics and accuracy in deciphering SVs suggest that it can be employed as a substitute for most other methods. Another advantage of this genome-wide technique is its ability to detect currently unknown variants, which might give insight into disease biology and refine risk stratification. In theory, these SVs might also be detectable by other methods such as FISH. However, designing a vast number of probes has not been feasible thus far. Furthermore, the benefits of using targeted methods to screen for novel variants are limited. Therefore, OGM, in comparison to other cytogenetic methods apart from WGS, can truly detect novel variants. Of course, these should be verified by other methods such as PCR or FISH. For the latter, the design of probes is more feasible, as long as one knows what to search for.

Although—for diagnostic purposes and to deliver results in a timely fashion—the samples in our study were analyzed with a focus on a predefined set of genes, genome-wide data are available, and detailed analysis will lead to the detection of novel variants. The same holds true for WGS; here, OGM might have the advantage of a quicker read-out, whereas a disadvantage might be the necessity for amplification-based techniques for the verification of novel findings. Thus, in this setting, OGM might serve as screening tool.

As an example, we detected a *DDX3X::MLLT10* fusion in a female AML patient. So far, this variant has only been described in rare male cases and mostly in T-ALL patients [38]. *MLLT10* is a common fusion partner of *KMT2A*. Interestingly, *DDX3X* escapes X inactivation, which led to the hypothesis that its variants might be leukemic driver mutations in male patients as they are mostly found in such a setting [39]. Besides *DDX3X::MLLT10* fusion, other aberrations, such as a *SUZ12* deletion, that are more common in ALL were detected by OGM in one such patient. Clinically, the course of disease was worse than expected. Due to an *FLT3*-ITD mutation, induction chemotherapy with cytarabine and daunorubicin was augmented with gilteritinib. However, the patient showed blast persistency after induction treatment. Salvage chemotherapy according to the FLAG-Eto regimen resulted in complete remission. In this case, OGM data led to the diagnosis of lineage ambiguity with not only AML-typical changes but characteristics of an early T-cell precursor ALL. Furthermore, the detected structural variants (*DDX3X::MLLT10* and the deletion of *SUZ12*) formed the basis for the hypothesis of a possible disease-enhancing mechanism via interaction with DOT1L and altered methylation [38].

In conclusion, these studies highlight how novel findings offer explanations for disease biology and have the potential to influence treatment decisions apart from mere risk stratification and classification. The characterization and functional validation of these findings will be the subject of future research. However, the genome-wide approach enables the acquirement of not only the information necessary for standard diagnostics but also additional data, which can provide a better understanding of this disease and its therapy.

## 9. Conclusions

In conclusion, OGM is a novel method for cytogenetic diagnostics. The easy access to blood and bone marrow without the need for any specific processing by a clinician after sample acquisition renders this method especially attractive for hematological malignancies with a leukemic disease course. One of OGM’s benefits is its unbiased genome-wide analysis of structural variants with high resolution. In a single workup, it can combine most of the current diagnostic yield of CBA, FISH, and CMA with substantial additional information regarding unseen structural variants and the clarification of unclear findings. Furthermore, OGM can detect unknown SVs. Its limitations are a lack of detection in so-called masked regions, which are mainly in the telomeric and centromeric regions of the chromosomes, and a certain flaw in detecting hyper- and hypodiploidy. This method’s turnaround time currently is at least one week, which is quicker than most established cytogenetic methods but probably slower than amplification-based techniques such as WGS at high-throughput centers. Compared to WGS, OGM is superior for the detection of large aberrations but cannot be used for SNV analyses.

In summary, the current data suggest a change in the genetic diagnostics applied to AML in the very near future. A next-generation diagnostic approach covering SVs and SNVs could lie either in the combination of genome-mapping approaches with WES or in WGS with additional targeted sequencing approaches.

## 10. Future Directions

The current data demonstrate the usefulness of OGM for the initial diagnosis of acute leukemias. A next step could be the undertaking of an evaluation regarding disease monitoring, especially if there is a case without SNVs, which could be followed up over the course of treatment as measurable residual disease. In these cases, OGM could be of some benefit, as almost all AML patients presented with SVs in the described studies, although OGM’s resolution was lower than that of other amplification-based techniques. Disease monitoring without the alteration of DNA through amplification or selection through cultivation also allows for the opportunity to obtain more data on disease evolution during treatment or in the case of recurrence. An automated approach to DNA preparation and labelling could improve turnaround-time. In the future, this approach and an improvement in the number of samples per run could render this method a true high-throughput technique.

## Figures and Tables

**Figure 1 cancers-15-01684-f001:**
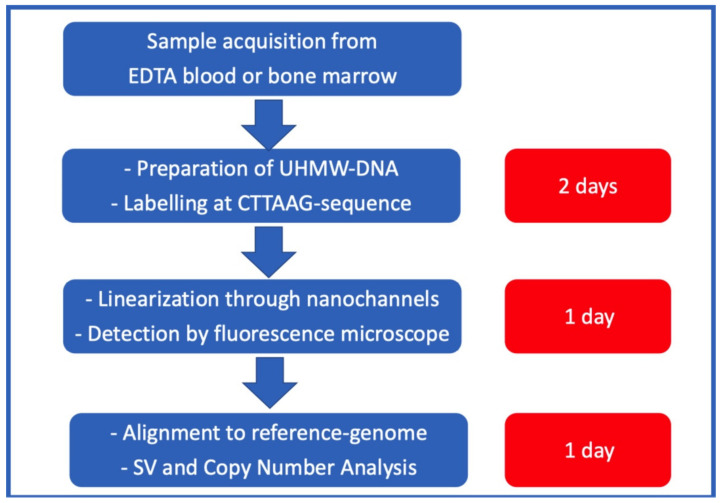
OGM workflow. After acquisition from blood or bone marrow, samples are either frozen or transferred to a lab. Ultra-high-molecular-weight (UMWH) DNA is prepared and labelled at CTTAAG sequence with a fluorochrome. The samples are transferred on a chip for linearization through nanochannels and run on a Bionano Saphyr^®^ system for detection by fluorescence microscopy. The genome is aligned according to a reference genome and analyzed through various algorithms with analysis of structural variants (SV) and according to copy number changes. Depicted timeline on the right shows minimum number of days required to analyze one sample.

**Figure 2 cancers-15-01684-f002:**
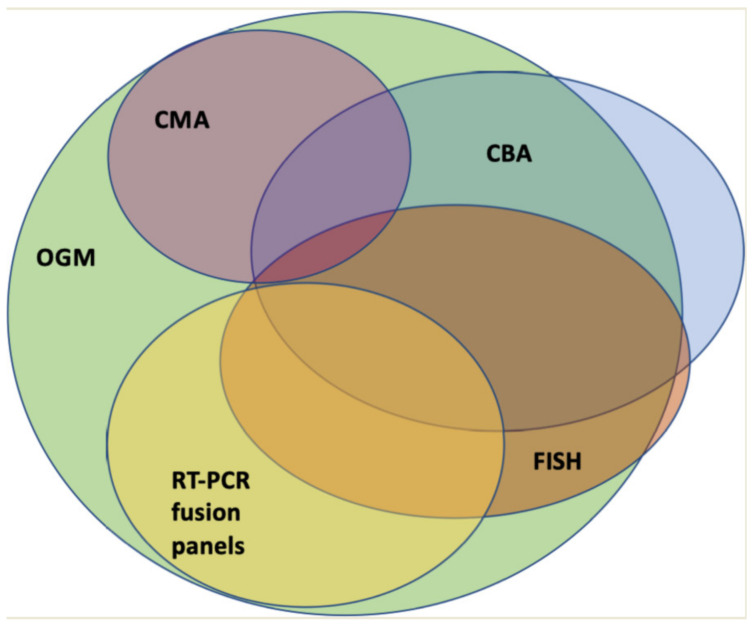
Schematic Graphical illustration of standard cytogenetic methods and OGM based on the data of Gerding et al. and Vangala et al. (quantitatively inaccurate) [27,28]. The vast majority of variants detected by conventional methods are visible by OGM, which, furthermore, enables the detection of a wider range of aberrations, i.e., novel variants, not visible by other techniques. However, a subset of variants, especially in highly repetitive regions such as telomeres and centromeres, might not be detected.

**Table 1 cancers-15-01684-t001:** (a): 2022 version of the European Leukemia Net risk stratification of AML [1]. Structural variants detectable by OGM are presented in bold. bZIP: Basic Leucine Zipper Domain. (b): Additional structural aberrations that are not part of ELN risk classification but are recurrent according to International Consensus Classification 2022, all of which are all detectable by OGM and, therefore, presented in bold.

(a)	
Risk Category	Genetic Variant
Favorable	**t(8;21)(q22;q22.1); *RUNX1::RUNX1T1*** **inv(16)(p13.1q22) or t(16;16)(p13.1q22); *CBFB::MYH11*** Mutated *NPM1* without *FLT3*-ITD bZIP in frame mutated *CEBPA*
Intermediate	Mutated *NPM1* and mutated *FLT3*-ITD Wildtype *NPM1* and mutated *FLT3*-ITD **t(9;11)(p21;q23.3)** **Cytogenetic abnormalities not classified as favorable or adverse**
Adverse	**t(6;9)(p23;q34.1) *DEK::NUP214*** **t(v;11q23.3) *KMT2A* rearranged** **t(9;22)(q34.1q11.2) *BCR::ABL1*** **inv(3)(q21.3q26.2) or t(3;3)(q21-3q26.2) *GATA2*, *MECOM* (*EVI1*)** **-5 or del(5q-); -7, -17/ abn(17p)** **Complex karyotype, monosomal karyotype** Mutated *ASXL1*, *EZH2*, *BCOR1*, *RUNX1*, *SF3B1*, *SRSF2*, *STAG2*, *U2AF*1, *ZRSR2* Mutated *TP53*
**(b)**	
**Additional structural aberrations**	
**t(15;17)(q24.1;q21.2)/*PML::RARA***	
**other *MECOM* rearrangements**	
**t(1;3)(p36.3;q21.3)/*PRDM16::RPN1***	
**t(1;22)(p13.3;q13.1)/*RBM15::MRTFA***	
**t(3;5)(q25.3;q35.1)/*NPM1::MLF1***	
**t(5;11)(q35.2;p15.4)/*NUP98::NSD1***	
**t(7;12)(q36.3;p13.2)/*ETV6::MNX1***	
**t(8;16)(p11.2;p13.3)/*KAT6A::CREBBP***	
**t(10;11)(p12.3;q14.2)/*PICALM::MLLT10***	
**t(11;12)(p15.4;p13.3)/*NUP98::KMD5A* or other *NUP98*r**	
**t(16;21)(p11.2;q22.2)/*FUS::ERG***	
**t(16;21)(q24.3;q22.1)/*RUNX1::CBFA2T3***	
**inv(16)(p13.3q24.3)/*CBFA2T3::GLIS2***	
**t(5q)/add(5q);del(7q);del(12p)/t(12p)/(add)(12p); i(17q);del(17p); del(20q); and/or idic(X)(q13)**	

**Table 2 cancers-15-01684-t002:** Overview of studies evaluating use of OGM for adult AML diagnostics. Due to the slightly different study designs and analyses employed, the classification of cases with “additional information” varies. Additional information consists of variants that were not detected by routine diagnostics or variants that were unclear and could be specified after routine diagnostics (CBA—Chromosomal Banding Analysis; FISH—fluorescence in situ hybridization; and CMA—CNV microarray) [21,22,27,28,30,31].

Study	Method	AML Cases	Concordance	Additional Information	Change in Classification/ Therapy
Neveling 2021 [21]	CBA, FISH, CMA	11	91%	-	-
Levy 2022 [30]	CBA, FISH, CMA	100	95% with CBA	13%	12%
Gerding 2022/ Vangala 2022 [27,28]	CBA, FISH, CMA	35	91%	64%	3%
Balducci 2022 [31]	CBA, FISH	41	88%	41%	5%
Sahajpal 2022 [22]	CBA, FISH, CMA	18	89%	33%	-

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
