# Peer review of "Optical Genome Mapping for Cytogenetic Diagnostics in AML"

_cancers, 2023, doi:10.3390/cancers15061684_

Round 1
Reviewer 1 Report
In this article, the authors review the methods currently used to diagnose genetic aberrations in acute myeloid leukaemia (AML), focusing on the new optical genome mapping (OGM) method.
In my opinion, due to the still limited usage of OGM in different genetic diagnostic areas and the context of hematological malignancies, it is helpful to provide an overview of the so-far available data.
Overall, the manuscript is well-written and contains scientifically valid, up-to-date information.
However, two major points need to be reviewed before considering the manuscript for publication:
-) The manuscript does not distinguish between the OGM and the described OGM platform, Saphyr, which gives the impression that the two are synonymous. The authors should clarify from the beginning that OGM, as discussed in this and presumably all other referenced studies, is limited to a specific commercial system.
-) Authors should be more precise about the term "additional information" when referring to the results of OGM. Are these additional aberrations not detected by other methods, or have they been detected by standard methods, but could they be characterized in greater detail by OGM? This information should be included to discern variants detected by OGM as a primary, stand-alone detection tool, exclusively capable and superior to other methods in making an aberration visible, and variants that could be further characterized by OGM as a downstream, complementary tool.
It should also be described why variants detected exclusively by OGM were undetectable by other means to make the exclusivity of the method more tangible to readers.
Besides this, there are a few minor points for consideration, as follows:
-) line 183: consider providing further details about Bionano´s “Rare-variant pipeline” (RVP) and its difference to the “de novo pipeline” (DNP). In the manuscript, the RVP is described as mapping “only the regions that are not matching the reference”, which is true for any region containing any structural variant.
-) line 188: “CNVs” (as defined in this manuscript and by Bionano) are not recognized by differences in label patterns, but by quantification of DNA-molecules
-) line 199: please provide the source of the stated detection limits of mosaicisms
-) line 424: please explain some of the more manufacturer-specific terms, e. g. “masked regions”
-) line 444: with the current capacity of Bionano’s Saphyr instrument, allowing a maximum throughput of 6 samples per 24 hours, the reviewer would not consider this a “high throughput” method
-) minor spelling corrections are needed (e.g. Fig. 1 “Detection”; “Bionano” is sometimes capitalized, and sometimes not; line 181: remove an extra period)
Author Response
Dear Reviewer,
Thank you for the kind review of our manuscript and your helpful comments.
Enclosed you will find the revised manuscript and the point by point response to your comments and suggestions with changes highlighted in yellow. We hope, we could address your concerns.
The manuscript does not distinguish between the OGM and the described OGM platform, Saphyr, which gives the impression that the two are synonymous. The authors should clarify from the beginning that OGM, as discussed in this and presumably all other referenced studies, is limited to a specific commercial system.
Thank you for this comment. In our initial version we have addressed this issue very briefly in line 168 ff. and in the legend to Figure 1.
We now have added following statements:
Line 40 ff: …we review current methods of cytogenetic diagnostics and summarize the knowledge on novel techniques with a focus on optical genome mapping (OGM) on the Saphyr system by Bionano Genomics (San Diego, USA)…
Line 159 ff: The method described here and the data summarized in this review was exclusively generated on the commercially available Saphyr system by Bionano Genomics (San Diego, USA). This also includes sample work up and data analysis on a platform provided by the company.
Authors should be more precise about the term "additional information" when referring to the results of OGM. Are these additional aberrations not detected by other methods, or have they been detected by standard methods, but could they be characterized in greater detail by OGM? This information should be included to discern variants detected by OGM as a primary, stand-alone detection tool, exclusively capable and superior to other methods in making an aberration visible, and variants that could be further characterized by OGM as a downstream, complementary tool. It should also be described why variants detected exclusively by OGM were undetectable by other means to make the exclusivity of the method more tangible to readers.
Additional information includes both: 1) the clarification and further characterization of unclear findings, mainly by CBA, such as unclear karyotypes and 2) the detection of aberrations missed by the combination of standard cytogenetic methods. Of course, the use of a wide range of FISH probes and CMAs in all patients would probably detect many of the “missed” aberrations. However, this is not the standard approach in most centers, also due to the high costs. For a variety of SVs, FISH probes either have to be designed or are available for research purposes, only, sometimes with a limited value as we ourselves had to experience. As mentioned in section 8 (OGM as a tool for detecting novel variants) we believe, that in some cases, OGM truly detects so far unknown aberrations, which vice versa can be verified by other methods.
We adapted the manuscript as following:
Line 250 ff: These studies primarily focused on the concordance of results regarding cytogenetic findings through OGM compared to routine diagnostics. Furthermore, all studies looked for additional information that could be acquired through OGM. This additional information was twofold: on the one hand unclear findings such as marker chromosomes or unclear complex karyotypes usually due to the low resolution of CBA could be resolved. On the other hand, FISH and CMAs were not utilized in all samples but were used as indicated by treating physicians and wherever established in routine diagnostics. Thus, additional information also included SVs detected by OGM, that in theory might be seen by methods such as FISH but far beyond routine diagnostics. As FISH-probes are usually only utilized for predefined aberrations, these variants would be missed by standard cytogenetic diagnostics.
Line 427 ff: In theory, these SVs might also be detectable by other methods such as FISH. However, designing a vast amount of probes has so far not been feasible. Furthermore, the benefit of targeted methods for screening for novel variants is limited. Therefore, OGM, in comparison to other cytogenetic methods apart from WGS, can truly detect novel variants. These of course, should be verified by other methods as PCR or FISH. For the latter, the design of probes is more feasible, when knowing what to look for.
Heading for table 2: Additional information consists of variants which were not detected by routine diagnostics or variants that were unclear and could be specified after routine diagnostics.
-) line 183: consider providing further details about Bionano´s “Rare-variant pipeline” (RVP) and its difference to the “de novo pipeline” (DNP). In the manuscript, the RVP is described as mapping “only the regions that are not matching the reference”, which is true for any region containing any structural variant.
-) line 188: “CNVs” (as defined in this manuscript and by Bionano) are not recognized by differences in label patterns, but by quantification of DNA-molecules
We adapted the paragraph starting in line 188 ff. (now lines 193ff) and added some more information regarding the RVP and DNP pipeline. We hope to have clarified the reviewer’s remark, although we are a little unsure regarding the comment for line 183.
We are thankful for the remark regarding CNVs and have changed this. The paragraph (lines 193ff) now reads:
The software offers two ways of analysis, the de novo-pipeline (DNP) and the rare-variant-pipeline (RVP). With the DNP, the DNA molecules are assembled to form a complete genome which in turn is compared to the reference genome (GRCh37/hg19 or hg38). The DNP offers a high the advantage of a high resolution but the limitation of missing aberrations with a low allele frequency. During RVP analysis, labelling patterns are aligned against the reference genome (GRCh37/hg19 or hg38). Differences of the aligned barcode pattern with the reference genome lead to the creation of a consensus genome map file (CMAP). The CMAP in turn is realigned with the reference genome, differences in alignment pattern are then called as SVs. Every aberration needs to be detected in 3 DNA molecules to be shown as such. The RVP is more commonly used for the analysis of hematological malignancies as it offers the possibility to detect mosaicism of aberrations. CNVs have a minimum length of 500 kb and are detected by the quantification of DNA molecules in comparison to the average number of all DNA molecules.
-) line 199: please provide the source of the stated detection limits of mosaicisms
Source has been provided (now line 229)
-) line 424: please explain some of the more manufacturer-specific terms, e. g. “masked regions”
We added details regarding the analysis pipelines as mentioned above. Furthermore we added the following:
line 232ff: Detection of SVs in centromeric and telomeric regions of the genome can be challenging. Here, coverage is limited due to high repetitive sequences and therefore lack of labelling (so called masked regions).
-) line 444: with the current capacity of Bionano’s Saphyr instrument, allowing a maximum throughput of 6 samples per 24 hours, the reviewer would not consider this a “high throughput” method
We agree with the reviewer; currently OGM is not a high throughput method. As stated in the last paragraph in regard to sample preparation. We addressed the reviewer’s comment in the last sentence of the manuscript as following.
line 490ff: This, and an improvement in samples per run could in the future render this method a true high-throughput technique.
The spelling mistakes have been corrected.
With kind regards, yours sincerely
Deepak B. Vangala,
Corresponding author
Center for hemato-oncological diseases
Knappschaftskrankenhaus
Ruhr-University Bochum, Germany

Reviewer 2 Report
The manuscript by Nilius-Eliliwi et al describes the advantages of a whole genome approach for detection of unbiased structural variants using optical genome mapping (OGM) in the diagnostic cytogenetic workup of AML. The comparison to current standard of care methods is clearly explained and recent literature is included. I have not comments other than that the cytogenetic resolution of GTG karyotyping is estimated to 5-10 Megabases. This is rather optimistic, since cytogenetic metaphases preparation from leukemia samples rarely allow a band quality above 400 bands, which is compatible to a resolution of > 10 Megabases. The standard cytogenetic resolution of 5-10 Mb described here corresponds to constitutional cytogenetic preparations.
Author Response
Dear Reviewer,
Thank you for the very kind and encouraging review of our manuscript and your helpful comments.
Enclosed you will find the revised manuscript and the point by point response to your comment with changes highlighted in yellow.
The manuscript by Nilius-Eliliwi et al describes the advantages of a whole genome approach for detection of unbiased structural variants using optical genome mapping (OGM) in the diagnostic cytogenetic workup of AML. The comparison to current standard of care methods is clearly explained and recent literature is included. I have not comments other than that the cytogenetic resolution of GTG karyotyping is estimated to 5-10 Megabases. This is rather optimistic, since cytogenetic metaphases preparation from leukemia samples rarely allow a band quality above 400 bands, which is compatible to a resolution of > 10 Megabases. The standard cytogenetic resolution of 5-10 Mb described here corresponds to constitutional cytogenetic preparations.
We have added the suggestion in line 104 ff:
Visual evaluation allows the experienced examiner to detect numerical and structural aberrations with a resolution limit of around 5-10 megabases (Mb) for constitutional cytogenetic preparations and > 10 Mb for leukemia samples
With kind regards, yours sincerely
Deepak B. Vangala,
Corresponding author
Center for hemato-oncological diseases
Knappschaftskrankenhaus
Ruhr-University Bochum, Germany

Reviewer 3 Report
This is a well-written review on the application of Optical Genome Mapping for genomic classification of acute myeloid leukemia.
I recommend the authors to address the following minor errors or the question which were raised during the review
1-Line 60: SXL1 should be corrected
2-Page 4, line 117: resolution of FISH is mentioned as 50bp which is not correct
3-Line 129-131: There is no citation for the possibility of FISH for whole genome analysis, also there is no continuity from this sentence to the next, can the authors rephrase them in order to make the message clear.
4- Page 5, line 188: please cite the reference for "CNVs have a minimum length of 500Kb". Also it should be mentioned based on which technology as this definition may change when more sensitive techniques are applied
5-Page 7, line 256-257: Would the author additional information to explain the weakness of OGM in calling ploidy changes?
Author Response
Dear Reviewer,
Thank you for the very kind and encouraging review of our manuscript and your helpful comments.
Enclosed you will find the revised manuscript and the point by point response to your comments and suggestions with changes highlighted in yellow.
I recommend the authors to address the following minor errors or the question which were raised during the review
1-Line 60: SXL1 should be corrected
We thank you for this correction.
2-Page 4, line 117: resolution of FISH is mentioned as 50bp which is not correct
We corrected this and added sources 10 and 11. It now reads:
line 118ff: A molecular expansion to CBA is fluorescence in situ hybridization (FISH). This targeted approach lowers the resolution threshold to about 100 – 200 kb in routine diagnostics and up to 5kb with specific alterations.
3-Line 129-131: There is no citation for the possibility of FISH for whole genome analysis, also there is no continuity from this sentence to the next, can the authors rephrase them in order to make the message clear.
References (Schröck et al, Science 1996; Speicher et al., Nature Genetics 1996) have been added (line 132). We rephrased the following sentence by adding “however” and hope to have sufficiently addressed the reviewer’s concerns. It now reads
line 132ff: For a whole genome analysis multicolor FISH is possible [12, 13]. However, due to the high costs because of the necessity of several probes for such questions, FISH is usually utilized in routine diagnostics for specific questions, only.
4- Page 5, line 188: please cite the reference for "CNVs have a minimum length of 500Kb". Also it should be mentioned based on which technology as this definition may change when more sensitive techniques are applied
We added the reference [20] and some more information regarding the analysis pipelines.
line 204ff: CNVs have a minimum length of about 500 kb and are detected by the quantification of DNA molecules in comparison to the average number of all DNA molecules. The platform for detection is the CN-analysis tool which is part of the DNP and the RVP.
5-Page 7, line 256-257: Would the author additional information to explain the weakness of OGM in calling ploidy changes?
This information is included in section 4 of the manuscript (lines 232 ff):
Detection of SVs in centromeric and telomeric regions of the genome can be challenging. Here, coverage is limited due to high repetitive sequences and therefore lack of labelling (so called masked regions). Apart from that, sometimes hyper- or hypodiploidy are missed. As the software calculates a relative number of gene copies, this is prone to mistakes if the entire genome has a hyper- or hypodiploid state.
With kind regards, yours sincerely
Deepak B. Vangala,
Corresponding author
Center for hemato-oncological diseases
Knappschaftskrankenhaus
Ruhr-University Bochum, Germany

Round 2
Reviewer 1 Report
The authors seem to have made a visibly superficial effort to improve the manuscript, and several points of my comments still need to be addressed.
The manuscript still does not to discern between the method of OGM and the proprietary system of Bionano Saphyr.
The language and phrasing of added passages are sometimes colloquial and faulty (e.g. “The DNP offers a high the advantage of a high resolution…”). Statements are not supported by argumentation (“These of course, should be verified by other methods…”).
I recommend addressing the comments with care and exhaustively.